# Acellular Human Amniotic Fluid-Derived Extracellular Vesicles as Novel Anti-Inflammatory Therapeutics against SARS-CoV-2 Infection

**DOI:** 10.3390/v16020273

**Published:** 2024-02-09

**Authors:** Debarati Chanda, Tania Del Rivero, Roshan Ghimire, Sunil More, Maria Ines Mitrani, Michael A. Bellio, Rudragouda Channappanavar

**Affiliations:** 1Department of Veterinary Pathobiology, Oklahoma State University, Stillwater, OK 74078, USA; debarati.chanda@okstate.edu (D.C.); roshan.ghimire@okstate.edu (R.G.); sunil.more@okstate.edu (S.M.); 2Organicell Regenerative Medicine, Davie, FL 33314, USA; tania.delrivero@gmail.com (T.D.R.); mari@dr-mari.com (M.I.M.)

**Keywords:** MA-CoV-2, acAF, AF-EVs, dysregulated immunity, cytokine storm, myeloid cells

## Abstract

The ongoing COVID-19 pandemic caused by SARS-CoV-2 is associated with acute respiratory distress syndrome (ARDS) and fatal pneumonia. Excessive inflammation caused by SARS-CoV-2 is the key driver of ARDS and lethal disease. Several FDA-approved drugs that suppress virus replication are in clinical use. However, despite strong evidence for the role of virus-induced inflammation in severe COVID-19, no effective anti-inflammatory drug is available to control fatal inflammation as well as efficiently clear the virus. Therefore, there is an urgent need to identify biologically derived immunomodulators that suppress inflammation and promote antiviral immunity. In this study, we evaluated acellular human amniotic fluid (acAF) containing extracellular vesicles (hAF-EVs) as a potential non-toxic and safe biologic for immunomodulation during COVID-19. Our in vitro results showed that acAF significantly reduced inflammatory cytokine production in TLR2/4/7 and SARS-CoV-2 structural protein-stimulated mouse macrophages. Importantly, an intraperitoneal administration of acAF reduced morbidity and mortality in SARS-CoV-2-infected mice. A detailed examination of SARS-CoV-2-infected lungs revealed that the increased protection in acAF-treated mice was associated with reduced viral titers and levels of inflammatory myeloid cell infiltration. Collectively, our results identify a novel biologic that has potential to suppress excessive inflammation and enhance survival following SARS-CoV-2 infection, highlighting the translational potential of acAF against COVID-19.

## 1. Introduction

Severe acute respiratory syndrome-coronavirus-2 (SARS-CoV-2), first identified in the human population in 2019, caused the COVID-19 pandemic [1]. Since its emergence, SARS-CoV-2 has infected over 700 million people and caused more than 6.9 million deaths worldwide [2]. SARS-CoV-2 primarily infects airway and lung epithelial cells, causing acute respiratory distress syndrome (ARDS), which is characterized by inflammation, acute lung injury (ALI), and, in many instances, progressing to fatal pneumonia [3,4]. Lethality in patients (especially the elderly and those with underlying co-morbidities) occurs due to high lung viral titers and an excessive influx of inflammatory monocyte/macrophages and neutrophils into the lungs [5,6,7,8]. Lung-infiltrating inflammatory myeloid cells further drive cytokine production during SARS-CoV-2 infection, leading to the development of the “cytokine storm”, a key feature of severe COVID-19 [7,9,10,11,12].

SARS-CoV-2 sensing by different immune sensors and the activation of downstream signaling pathways that induce antiviral and inflammatory cytokine/chemokine production has been well established [13,14,15,16]. Among these sensors, Toll-like receptor 7 (TLR7) and melanoma differentiation-associated protein 5 (MDA5) are the primary sensors of SARS-CoV-2 single-stranded RNA (ssRNA) and double-stranded RNA (dsRNA), respectively, while TLR2 and TLR4 detect viral envelope and spike/nucleocapsid proteins [13,15,16,17,18,19,20,21,22,23]. The activation of these sensors induces a cascade of inflammatory signaling via the tumor necrosis factor receptor-associated factor 6 (TRAF6), as well as the antiviral interferon response, primarily through TRAF3 [24]. Additionally, SARS-COV-2 ORF3a, envelope, and nucleocapsid proteins are sensed by the NLRP3 inflammasome [25,26,27]. Impaired antiviral IFN/ISG responses and an enhanced expression of pro-inflammatory cytokines, such as tumor necrosis factor (TNF), interleukin 6 (IL-6), and IL-1β, are observed in the lung tissues of patients with a lethal SARS-CoV-2 infection [4,8,11,28,29]. Therefore, the identification of a potential therapeutic that inhibits excessive inflammatory responses and promotes anti-viral immunity against SARS-CoV-2 infection is essential for moderating inflammation and host protection.

Several FDA-approved drugs that suppress virus replication are currently in clinical use for COVID-19 treatment. Novel anti-viral drugs such as remdesivir improve clinical outcomes in hospitalized COVID-19 patients [30]. Lagevrio, comprising molnupiravir, a viral nucleoside analog, was authorized for emergency use (EUA) by the FDA for the treatment of mild-to-moderate COVID-19 in adults (EUA 108 Merck Lagevrio LOA 02012023) [31,32,33], while Paxlovid, comprising a SARS-CoV-2 and HIV-1 protease inhibitor (nirmatrelvir and ritonavir, respectively), was authorized by the FDA for treating both adults and pediatric COVID-19 patients (EUA 105 Pfizer Paxlovid LOA 02012023) [34,35]. Additionally, recombinant IFNs administered during the initial stages of infection mildly alleviated disease symptoms by increasing viral clearance and reducing lung inflammation [36,37,38,39]. However, IFN therapy was not effective when administered in later infection stages as it increased mortality [40,41,42]. Anti-cytokine/inflammatory drugs such as tocilizumab (anti-IL-6 receptor-blocking antibody), anakinra (anti IL-1β antibody), baricitinib (Janus kinase1/2 inhibitor), and corticosteroids (such as dexamethasone) have been administered in several clinical trials to ameliorate hyper-inflammatory responses [43,44,45,46,47,48]. However, incomplete protection, as well as the ability of some of these anti-inflammatory agents to dampen antiviral IFN/ISG and adaptive immune responses, has limited their clinical success [49,50]. Therefore, a novel and safe anti-inflammatory and anti-viral drug with limited cytotoxicity is critically needed.

Extracellular vesicle (EV)-based therapies are emerging as the next generation of cell-derived biologics for a variety of inflammation-related diseases, including respiratory, cardiovascular, neurological, and autoimmune disorders [51,52,53,54]. EVs are cell-derived lipid-enclosed acellular nanoparticles that play an important role in intercellular communication [55,56,57,58]. EVs serve as carriers of proteins, lipids, nucleic acids, and metabolites, and transport anti-inflammatory and immunomodulatory signals to diverse cellular targets [59,60,61,62]. They are also involved in several biological processes such as immunomodulation, cellular proliferation/differentiation, coagulation, and senescence, to name a few [63,64,65,66,67,68,69]. Acellular amniotic fluid (acAF), which represents the natural secretome of extracellular factors secreted by the fetus and perinatal tissue, contains high concentrations of therapeutic and anti-inflammatory EVs rich in miR-146a and thioredoxin-1 [51,70]. These EVs can be harnessed for therapeutic use and have been shown to have anti-inflammatory and tissue-regenerative properties in pre-clinical models of inflammatory lung injury [51]. Furthermore, in a recent expanded-access, proof-of-concept clinical trial conducted by our group, the administration of an acAF biologic containing AF-EVs called Zofin was found to be safe, along with providing improved clinical symptoms and a reduction in inflammatory biomarkers [71]. Consequently, acAF and AF-EVs can be used to moderate acute inflammation and lung injury occurring in patients with COVID-19. Due to its acellular origin, the therapeutic potential of acAF is potentially derived from AF-EVs, along with other paracrine factors present in full-term AF [71]. Based on this previously published evidence, we hypothesized that acAF imparts protective anti-inflammatory and immunomodulatory effects by decreasing pro-inflammatory cytokines and increasing the anti-viral IFN response, confirming their translational potential.

Here, we examined the therapeutic potential of acAF in controlling SARS-CoV-2-induced excessive inflammation and in providing protection from lethal pneumonia. We found that acAF reduced viral titers in vitro in A549-hACE2 cells infected with SARS-CoV-2. Additionally, acAF and AF-EVs inhibited inflammatory cytokine production by TLR and SARS-CoV-2 structural protein-stimulated macrophages. Our in vivo therapeutic studies using a mouse model of SARS-CoV-2 showed that acAF provided significant protection from lethal infection, and this protection correlated with reduced lung viral titers, inflammatory monocyte–macrophages, and neutrophil influx in the lungs. Collectively, we determined that acAF treatment alleviates the inflammatory response and high viral replication induced by SARS-CoV-2 infection and provides protection, thereby ensuring recovery from severe disease.

## 2. Materials and Methods

### 2.1. Cells and Virus

The human airway epithelial A549-hACE2 (A549-hACE2 obtained from BEI resources, ATCC: 53821) and VeroE-hACE2 cells (BEI resources, ATCC: NR53826) were grown in T75 and T175 flasks (Corning, New York, NY, USA) using complete Dulbecco’s Modified Eagle’s Medium (DMEM, Gibco, Waltham, MA, USA, catalog #12100-038) supplemented with 10% fetal bovine serum (FBS), 1% penicillin–streptomycin, 1% l-glutamine, 1% sodium bicarbonate, 1% sodium pyruvate, and 1% non-essential amino acids. A mouse macrophage cell line (RAW 264.7, ATCC: TIB-71) was grown in T75 flasks (Corning) using HyClone RPMI-1640 (Cytiva, Marlborough, MA, USA, catalog #SH30027.02) supplemented with 10% fetal bovine serum (FBS), 1% penicillin–streptomycin, 1% l-glutamine, and 1% HEPES.

The WT SARS-CoV-2 Washington isolate (BEI resources, USA-WA1/2020, NR-52281) was used for all in vitro infections. SARS-CoV-2 (MA-CoV-2), developed by the serial passage of an N501Y mutant SARS-CoV-2 in mouse lungs provided by Dr. Stanley Perlman (U Iowa), was used for all in vivo infections. Details of the MA strain of SARS-CoV-2 (MA-CoV-2) are described in published studies [72,73].

### 2.2. acAF Collection, Preparation, and Cytotoxicity Evaluation

acAF is an allogenic, acellular biologic containing AF-EVs and other soluble extracellular factors naturally occurring in full-term human amniotic fluid (Organicell Regenerative Medicine, Inc., Miami, FL, USA). The acAF used was collected and prepared as previously described [71,74]. Donors were qualified under FDACFR 1271 guidelines and were certified following a review of their medical history, social history, physical examination, and raw product extraction information. Part of the donor qualification program included answering yes/no to a question on whether donors had had COVID-19 during their pregnancy. Only donors responding no were accepted for acAF collection. In short, human AF donated from consenting adults during routine, planned cesarean sections under IRB-approved donor screening (IRB approval agency: IRCM) was centrifuged and filtered to create the acellular product specified. In this study, we used 3 different lots of acAF (hereafter referred to as lots 1, 2, and 3) derived from 3 different human donors. Working solutions of acAF were prepared in phenol red-deprived DMEM (Gibco, catalog #31053-028) and RPMI 1640 (Gibco, catalog #11835-030) supplemented with 2% FBS, 1% penicillin–streptomycin, 1% l-glutamine, 1% sodium bicarbonate, 1% sodium pyruvate, 1% non-essential amino acids, and 1% HEPES, respectively. To test the cytotoxic effect of acAF on A549-hACE2 cells and RAW 264.7 cells, a Lactate Dehydrogenase (LDH) cytotoxicity assay kit (Cell Biolabs, Inc., San Diego, CA, USA, CytoSelect^TM^ LDH Cytotoxicity Assay kit catalog #CBA-241) was used.

### 2.3. Nanoparticle Tracking Analysis of acAF Preparations

A nanoparticle tracking analysis of the acAF preparation was performed using a NanoSight NS300 instrument and software (Malvern Panalytical, Malvern, UK, NTA 3.4 Build 3.4.003). A total of 10 µL of the sample was diluted in 10 mL of cell culture-grade water. The capture settings were modified to capture 5 video files with a capture duration of 30 s, using a camera level of 15 and a continuous syringe pump flow rate of 50. After the completion of the script, the video files were analyzed with a detection threshold of 3. The NTA post-acquisition settings were kept constant between samples. Fluorescent NTA (fNTA) was performed using a ZetaView QUATT with the ZetaView software, version 8.05.12 SP1 (Particle Metrix GmbH, Inning am Ammersee, Germany). acAF nanoparticles were labeled with anti CD81-DyLight 550 (NB100-65805R) (Novus Biologicals, Littleton, CO, USA) by adding 1 mL of each fluorescent antibody to 20 mL of the sample containing isolated EVs. The fluorescently labeled EV samples were then incubated for 2 h in the dark on ice. The samples were diluted by mixing deionized water filtered through a 0.2 µm syringe filter with corresponding volumes of the sample. The fNTA was performed in scatter mode and 520/550 fluorescent mode. For scatter mode analysis, the ZetaView settings were adjusted to have a sensitivity of 75, a shutter speed of 100, cycles/positions of 2/11, a frame rate of 30, a maximum size of 1000, a minimum size of 20, a track length of 15, and a minimum brightness of 20. The fluorescent-mode analysis had similar parameters except for an increased sensitivity of 8085. The size and concentration profiles of each mode were then imported into Prism (Graph Pad, version 9.2.0) and superimposed (Figure 1A,B). This experiment was repeated with the AF-EV preparations from each acAF.

### 2.4. AF-EV Preparation

AF-EVs were precipitated from acAF by ultracentrifugation. The acAF was spun at 100,000× *g* for 90 min in the XPN-90 Ultracentrifuge (Beckman Coulter, Brea, CA, USA) with the 50.4 Ti rotor at 4 °C. The resultant nanoparticle pellets were re-suspended with 1 mL of 0.9% normal saline (B. Braun). The nanoparticle concentration of the resuspended product was completed, and the samples were adjusted to nanoparticle concentrations of approximately 2.0 × 10^10^ nanoparticles per 100 µL.

### 2.5. Cytokine Array Dot Blot Analysis

The acAF was characterized using the human cytokine antibody array (membrane, 120 targets (ab193656)). A total of 1 mL of acAF was incubated on the membranes overnight at 4 °C. The next day, the membranes were washed and incubated with streptavidin and HRP secondary antibodies. Blots were developed using the Bio-Rad ChemiDocs Touch imaging system (Bio-Rad, Hercules, CA, USA). The images were saved and the spot intensity was quantified using the Image Lab software. The relative expression of the proteins was determined by first subtracting the intensity value for the negative and blank spots. The average value for each protein was calculated by dividing the intensity value of the protein by the average intensity value of the positive control spots from the respective blot. The relative expression of each protein was plotted to show the high and low detected proteins. This analysis was completed with 3 different lots of acAF. The relative intensity of each cytokine blot was imported into Prism (Graph Pad, version 9.2.0) and added (Figure 1C,D).

### 2.6. In Vitro Virus Infection and acAF Treatments

A confluent monolayer of A549-hACE2 cells seeded in 24-well plates were treated with 3 different lots of acAF at concentrations of 5% and 25% *v*/*v*, starting 2 h (hr) before SARS-CoV-2 infection. The control D-2 media consisted of DMEM supplemented with 2% FBS, 1% sodium bicarbonate, 1% sodium pyruvate, and 1% non-essential amino acids, and the acAF pre-treated cells were infected with a 0.1 multiplicity of infection (MOI) dose of wild-type (WT) SARS-CoV-2. At 16 h and 48 h post-infection, the cell supernatants were collected for a virus titer assay, and the cells were lysed with a Trizol reagent (Life Technologies, Carlsbad, CA, USA, catalog #15596018) and stored at −80 °C. Small aliquots of the clarified cell supernatant were titered on VeroE-hACE2 cells using a standard plaque assay protocol [75,76]. The Trizol-lysed cells were later used to evaluate the expression of anti-viral genomic and sub-genomic RNA levels.

### 2.7. In Vitro Ccytokine and Chemokine Analysis upon Stimulation with SARS-CoV-2 Structural Proteins and TLR Agonists

RAW 264.7 macrophage cell lines were seeded in a 48-well plate, and a confluent monolayer of cells was pre-treated for 2 h with 3 different lots of acAF and AF-EVs at concentrations of 25% *v*/*v* and 10% *v*/*v*, respectively, in 1 mL solutions. Following treatment, the cells were stimulated with Toll-like receptor agonists PAM3CSK4 (TLR1/2 agonist, InvivoGen, catalog # vac-pms) at 1 µg/mL, LPS (TLR4 agonist, InvivoGen, catalog # tlrl-smlps) at 100 ng/mL, and R837 (TLR7 agonist, InvivoGen, catalog # tlrl-imq) at 1 µg/mL. In another study, acAF-treated RAW 264.7 macrophages were stimulated with recombinant SARS-CoV-2 structural proteins—spike (S) at 5 µg/mL (BEI Resources, catalog #NR52397), envelope (E) at 0.5 µg/mL (ABclonal, catalog # RP01263LQ), and membrane (M) at 1 µg/mL (M, Abcam, catalog # 48951) proteins for 12 h. The collected supernatant was diluted to 1:32 followed by cytokine estimation using an ELISA. The Mouse TNF ELISA Set II (BD OptEIA, catalog # 558534) was used for performing the TNF ELISA.

### 2.8. Mice Infection and acAF Treatments

BALB/C mice (10–15 weeks old, female, procured from Charles River, Strain Code # 028) were injected intra-peritoneally (I.P) with 2 different lots of acAF (lot 2 and lot 3) at 0 and 2 days post-infection (DPI) at a concentration of 200 µL (100 µL acAF + 100 µL PBS)/mouse. One single dose of lot 3 contained approximately 4.04 × 10^10^ nanoparticles, whereas 1 single dose from lot 2 contained approximately 2.28 × 10^10^ nanoparticles (Figure 1A). An equal amount (200 µL) of 1× phosphate-buffered saline (PBS) was injected intra-peritoneally (I.P) in control mice. The control PBS and acAF-treated mice were infected with 250 PFUs/mouse of MA-CoV-2 via the intra-nasal route (I.N.) in 50 µL of DMEM (vehicle, Gibco, catalog #12100-038) under isoflurane (Pivetal, Patterson Veterinary) anesthesia. For studying morbidity and mortality post-infection, the mice were weighed and examined daily for 12 days to assess clinical signs such as their respiratory rate, posture/movement, fur condition, and ability to eat and drink [75]. A 25% loss of the initial body weight or signs of severe pneumonia (clinical illness) were considered as an endpoint, and mice with a severe disease score or >25% initial body weight loss were humanely euthanized by isoflurane overdose. All procedures involving animals were approved by the Oklahoma State University Animal Care and Use Committee under the American Veterinary Medical Association guidelines. To harvest lung tissues, groups of mice were euthanized at 4 DPI using an isoflurane overdose followed by cervical dislocation. The lung tissues were PBS-perfused and divided into two parts (right and left lobes). One part was consistently used for analyzing viral titers and gene expression, while the other part was used for the immuno-labeling of myeloid cells.

### 2.9. Virus Titer Assay and Gene Expression

Lung tissues (right lobe) harvested in PBS were homogenized using a mechanical bead homogenizer (Bead Mill4, Fisher Brand) and divided into two equal parts. One part of the lung homogenate was freeze–thawed, and 200 µL of a 10-fold serially diluted sample was titered on VeroE-hACE2 cells seeded in a 12-well plate. The serially diluted and infected Vero-hACE2 plates were incubated at 37 °C in a 5% CO_2_ cell culture incubator with brief rocking every 10 min. The virus inoculum was removed after 1 h and the wells/cells were washed with 500 µL of PBS to remove any un-bound virus. The cell monolayer was overlayed with 1 mL of a 1:1 mixture of 1.2% agarose and 2× DMEM (2× DMEM powder, 2% FBS, 1% penicillin–streptomycin, 1% l-glutamine, 1% sodium bicarbonate, 1% sodium pyruvate, 1% non-essential amino acids). The plates were incubated in a 37 °C/5% CO_2_ cell culture incubator for 72 h. After 72 h of incubation, the cells were fixed with 10% paraformaldehyde for 30 min, followed by agarose removal and staining with 250 µL of crystal violet (0.1%) for 10 min. The crystal violet stain was removed and the cells were washed with PBS followed by plaque counting.

The other part of lung homogenate was lysed in 1 mL of a Trizol reagent and used for evaluating the mRNA levels of inflammatory and antiviral cytokines and viral genomic RNA levels using qPCR. RNA was extracted from the Trizol-lysed tissue, followed by cDNA synthesis (M-MLV Reverse transcriptase, Carlsbad, CA, USA, cat #2458204) and qPCR (PowerUp SYBR Green Master Mix, Applied Biosystems, Foster City, CA, USA, catalog #A25742), to estimate the expression of inflammatory cytokines/chemokines and anti-viral genes.

### 2.10. Immuno-Labeling and Fluorescence-Activated Cell Sorting (FACS) Assay to Assess Llung Myeloid Cells

For analyzing lung-infiltrating myeloid cells, PBS-perfused lung tissue (left lobe) was treated with a collagenase-D- and DNAse-1-containing tissue digestion buffer. Isolated cells were surface-immunolabeled for inflammatory monocytes (CD45^+^ CD11b^+^ Ly6C^hi^) and neutrophils (CD45^+^ CD11b^+^ Ly6G^hi^) and analyzed by FACS Aria-II flow cytometry. Briefly, ~5 × 10^5^–1 × 10^6^ lung cells were incubated with an Fc block (anti-CD16/32) for 10 min on ice. Following Fc blocking, the cells were washed with the FACS buffer (1× PBS, 2% FBS, and 0.05% sodium azide). Cell surface staining was then performed by labeling total lung cells with the following fluorochrome-conjugated monoclonal antibodies: PECy7 α-CD45 (clone: 30-F11); BV421 α-CD11b (clone: M1/70); PerCp-Cy5.5 α-Ly6C (clone: HK1.4); and FITC α-Ly6G (clone: 1A8) (all from BioLegend, San Diego, CA, USA, unless otherwise stated). Details of the cell surface immuno-labeling protocol for flow cytometry were described by Channappanavar et.al., 2020 [77]. A final concentration of 1:200 (antibody/FACS buffer) was used for all fluorochrome-conjugated antibodies, except for PerCp-Cy5.5-labeled antibodies used at a 1:300 concentration.

### 2.11. Data Analysis

A statistical analysis was performed using GraphPad Prism Version 8.0 (GraphPad Software Inc., San Diego, CA, USA). The results were analyzed using a one-way ANOVA. Data in scatter plots or bar graphs are represented as the mean ± standard error of the mean (SEM). Survival curves were assessed for statistical significance using a log-rank (Mantel–Cox test) or Gehan–Breslow–Wilcoxon test. Values of *p* < 0.05 were considered statistically significant.

## 3. Results

### 3.1. acAF Preparations Contain AF-EVs and Cytokines/Chemokines and Are a Non-Toxic Biologic

The characterization of the acAF preparation and AF-EV component has been previously published [71,74]. The three product lots used in the following experiments had a mean nanoparticle concentration of 2.06 × 10^11^/mL, with an 87.4 nm mode size (Figure 1B). The fluorescent nanoparticle tracking analysis determined that the nanoparticles within the acAF preparations were, on average, 63.6% positive for the EV marker CD81 (Figure 1A,B). hAF-EVs within acAF may play a protective role against SARS-CoV-2 infection due to their ability to inhibit hyper-reactive host immune responses such as inflammation [71]. A high expression of the tissue inhibitor of matrix metalloproteinase-2 (TIMP-2) cytokine was observed (Figure 1C,D) in the dot blot density-analyzed acAF samples. Considering the potential role of TIMP-2 in indirectly suppressing TNF induction, acAF is a good immunomodulatory candidates for use during SARS-CoV-2 infection [78]. Here, to assess the safety and toxicity of acAF, we first examined whether the acAF preparation used in our study was cytotoxic to A549-hACE2 (lung epithelial cell line) and RAW 264.7 (murine macrophage cell line) cells. We found that the LDH release was comparable in the control (media-only, negative control), 5%, and 25% concentrations of acAF-treated A549-hACE2 cells. The LDH release observed in the lot 1 (5% and 25%), lot 2 (25%), and lot 3 (25%) treated cells were below detection limits. Relative cytotoxicities of 3.5% in lot 2 (5%) and 0.8% in lot 3 (5%), which are both well below the acceptable cytotoxicity range of EC10, were observed (Figure 1E). In RAW cells, however, we found approximately 14% and 6% relative cytotoxicity in lot 2 (5%) and lot 3 (5%), respectively, while the LDH release for the remaining concentrations of lots 1, 2, and 3 were below the detection limits (Figure 1F). These results establish acAF as non-toxic and safe biologic in human lung epithelial and mouse macrophage cell lines.

**Figure 1 viruses-16-00273-f001:**
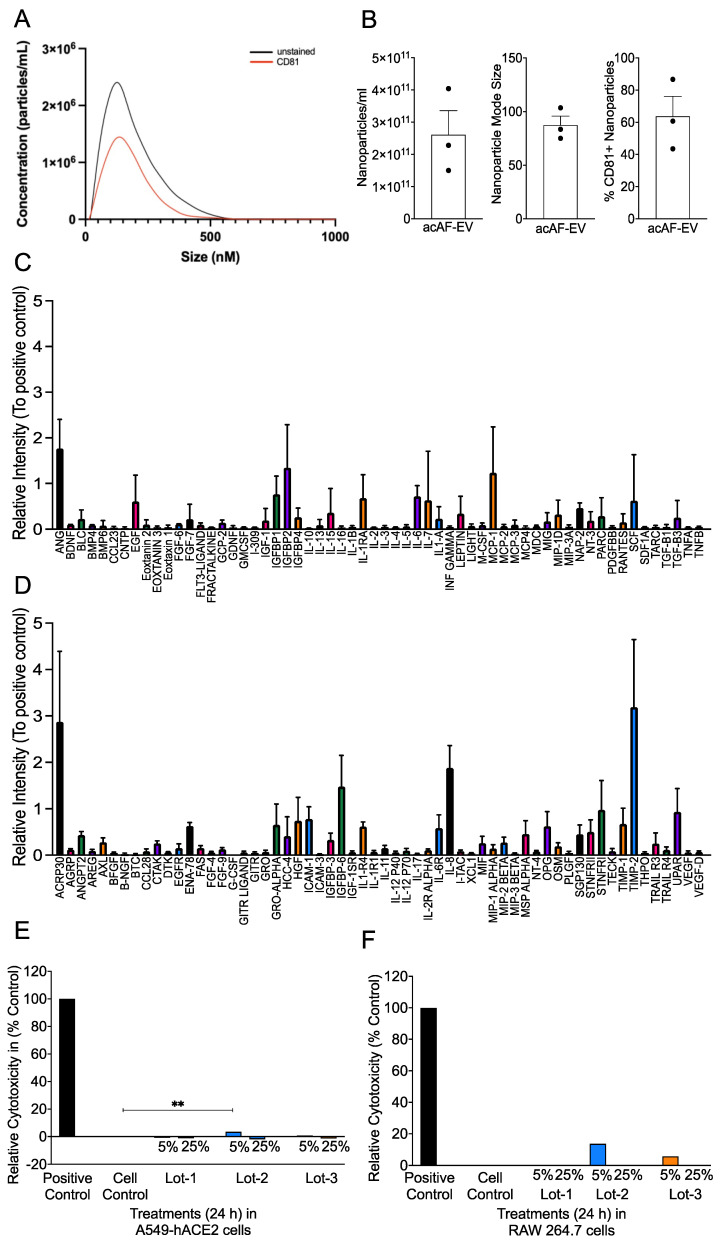
acAF preparations contain AF-EVs, cytokines, and chemokines, and are non-toxic biologics. acAF was analyzed for nanoparticle concentration and cytotoxicity against A549-hACE2 and RAW 264.7 cells. (**A**) Nanoparticle tracking analysis of the acAF preparations revealed representative nanoparticle size distribution of the total nanoparticles compared to CD81+ nanoparticles; average nanoparticle concentration distribution of the acAF preparations, N = 3; (**B**) Average nanoparticle concentration and mode size within the acAF preparations, N = 3; and percentage of nanoparticles positive for CD81 within the acAF preparations, N = 3. (**C**,**D**) Quantification of the relative intensity for each analyzed cytokine blot in dot blot (ID#C6, abcam) relative to positive control. (**E**,**F**) Supernatant collected from A549-hACE2 and RAW 264.7 cells treated with 5% and 25% concentrations of acAFs were examined for cytotoxicity. LDH release was analyzed to determine relative cytotoxicity (% of control). acAFs were found to be non-toxic to both cell types. Bar graphs show relative cytotoxicity (percentage of control). Data are representative of 2–3 independent experiments and represented as ±SEM (**E**,**F**). Statistical significance was determined using one way ANOVA, post hoc Dunnett’s test with ** *p* < 0.01.

### 3.2. acAF Reduces Virus Titers and Promotes Anti-Viral Response to SARS-CoV-2 Infection

In order to examine if acAF has antiviral properties, we assessed SARS-CoV-2 titers in acAF-treated A549-hACE2 cells. In these studies, we used the same acAF lots previously tested (in the above section), each at 5% and 25% concentrations. A549-hACE2 cells were infected with SARS-CoV-2 (0.1 MOI), and cell supernatants collected at 16 h and 48 h post-infection were used to titer on VeroE-hACE2 cells. DMEM-2% (D-2) was used as the control medium for infection studies. Our virus titer assays showed an approximately two–three-fold reduction in SARS-CoV-2 titers in lot 3 (5%) at 16 h post-infection, and in lot 1 (5% and 25%) and lot 3 (5% and 25%), at 48 h post-infection in acAF-treated cells when compared to the control D-2-treated cells (Figure 2A,B). These results demonstrate that acAF suppresses SARS-CoV-2 infection in A549-hACE2 cells. Next, we evaluated whether acAF suppresses virus replication through the robust induction of antiviral cytokines. A549-hACE2 cells were pre-treated for 2 h with different lots of acAF at concentrations of 5% and 25%. The cells were then infected with a 0.1 MOI dose of wild-type (WT) SARS-CoV-2. At 16 h post-infection, the cells were lysed with a Trizol reagent to isolate RNA and prepare cDNA, followed by qPCR for estimating the mRNA levels of antiviral genes. We found that the treatment with lot 3 (25%) caused a significant increase in anti-viral (IFNβ) mRNA levels (Figure 2C) at 16 h post-infection. These results suggest that acAF-induced anti-viral IFN effects suppress SARS-CoV-2 infection in lung epithelial (A549-hACE2) cells.

### 3.3. acAF and AF-EVs Suppress Inflammatory Cytokine Production in TLR Agonist- and SARS-CoV-2 Structural Protein-Stimulated Mouse Macrophages

The SARS-CoV-2 virion contains four structural proteins (S, E, N, and M) that are recognized by different TLRs and induce a strong inflammatory cytokine/chemokine response [13,17,79]. Several synthetic molecules called TLR agonists, such as PAM3CSK4 (TLR1/2 agonist), LPS (TLR4 agonist), and R837 (TLR7 agonist), can mimic microbial signals and stimulate TLR activation, leading to pro-inflammatory cytokine induction [80,81,82,83]. TLR-mediated inflammation is one of the major phenomena driving SARS-CoV-2-associated pathogenicity and disease severity [22,84]. Our published studies, as well as other studies, further showed that monocyte–macrophages are highly pro-inflammatory and key drivers of CoV-induced pathogenic lung inflammation [75,76,77,85]. Therefore, to assess whether acAF and AF-EVs suppress SARS-CoV-2-induced inflammation, we stimulated mouse macrophage cells (RAW 264.7) with SARS-CoV-2 structural proteins—spike (S), envelope (E), and membrane (M)—and TLR2, 4, and 7 agonists. AF-EVs were isolated from acAF and used to separately test their effects against TLR-agonist and SARS-CoV-2 structural proteins. RAW 264.7 cells were pre-treated with acAF (25% *v*/*v*) and AF-EVs (10% *v*/*v*) for 2 h, followed by the stimulation of cells with S, E, and M proteins; an ssRNA mimic (R837); and TLR2 and TLR4 agonists (PAM3CSK4 and LPS). Cell supernatants, collected 12 h post-stimulation, were used to assess the protein levels of TNF, a representative inflammatory cytokine. We observed that all lots of acAF reduced the TNF protein levels in the cell cultures stimulated with all three TLR agonists and E and M recombinant structural proteins (Figure 3A,B) compared to TLR agonist- and recombinant protein-only-stimulated control cells by more than two-fold. For the AF-EV stimulations, we found that all lots of AF-EVs reduced TNF levels in TLR2, E- and M-protein-stimulated cells (Figure 3C,D) compared to TLR agonist- and recombinant protein-only-stimulated control cells by approximately 1.5- to 2-fold. Collectively, these results demonstrate an anti-inflammatory effect of acAF or AF-EV treatment in mouse macrophages stimulated with TLR agonists and recombinant SARS-CoV-2 structural proteins.

### 3.4. acAF Provides Protection against SARS-CoV-2-Induced Lethal Disease and Mortality

Given the ability of acAF to suppress inflammatory cytokine production, we then examined the protective role of acAF in 10–15-week-old female BALB/C mice treated with two different lots of acAF (lots 2 and 3) and infected with MA-CoV-2. acAF lots 2 and 3 were selected because of their superior efficacy in reducing inflammatory cytokines and viral replication in vitro, as shown in Figure 3A,B. To assess the role of acAF in host protection, we injected 10–15-week-old mice intraperitoneally (I.P.) with the selected lots of acAF at 0 and 2 days post-infection (DPI) at a concentration of 200 µL (100 µL acAF + 100 µL PBS)/mouse. An equal amount (200 µL) of 1× phosphate-buffered saline (PBS) was injected I.P in the control mice. The control PBS- and acAF-treated mice were infected with 250 PFUs/mouse of MA-CoV-2 intranasally (I.N.) in 50 µL of DMEM. We found that the mice treated with both lots of acAF initially lost around 15% of their body weight and then started gaining weight from day 6 post-infection, whereas the control PBS-treated mice lost more than 25% of their body weight and all mice in this group succumbed to infection by day 7 post-infection (Figure 4A). The body weight loss was lower in the lot 2 and lot 3 acAF-treated mice compared to the control PBS-treated mice at the 6 DPI time-point (Figure 4A). The acAF-treated mice recovered almost 90% of their initial body weight by 12 DPI. Additionally, 60% of the lot 2-treated mice and 80% of the lot 3-treated mice survived to 8 DPI compared to the 100% mortality observed in the control PBS-treated mice by 7 DPI (Figure 4B). Next, to test whether the acAF treatment would reduce SARS-CoV-2 titers in the lungs, we examined the virus titers in the control and acAF-treated lungs at 4 dpi. Our results showed that the lot 3 acAF-treated mice had a two-fold reduction in viral titers compared to the control and lot 2 acAF-treated mice (Figure 4C). From these results, we conclude that acAF lot 3 is more effective as a therapeutic than lot 2. Collectively, these results demonstrate the protective immunomodulatory and anti-viral properties of acAF against MA-CoV-2 infection, as well as the ability of acAF to moderately suppress viral titers in MA-CoV-2-infected mice lungs and improve survival rates.

### 3.5. acAF Suppresses Myeloid Cell Accumulation in Mouse Lungs

Perinatal-derived products have been previously investigated for their potential to inhibit the chemotactic activity and influx of inflammatory myeloid cells such as monocytes, macrophages, and neutrophils [86,87,88]. Previous studies have shown that an excessive influx of inflammatory monocyte–macrophages and neutrophils induces an exuberant inflammatory response, which can proceed to a cytokine storm [4,75,76,77,89]. This acute inflammation can lead to fibrotic complications, multi-organ system damage, and death [10,90,91]. Since the excessive infiltration of inflammatory monocyte–macrophages (IMMs) and neutrophils in the lungs is a key driver of severe SARS-CoV-2 infection [92,93,94,95,96], we wanted to study whether acAF can suppress the accumulation of inflammatory myeloid cells in the lung tissues of MA-CoV-2-infected mice. PBS-perfused lung tissues harvested from mice at 4 DPI were analyzed for lung-infiltrating IMMs and neutrophils by FACS immuno-labeling. We observed a significant reduction in the levels of total IMMs (Figure 5C) and neutrophils (Figure 5F) in the lungs of infected mice treated with acAF lot 3 compared to the control (PBS-treated). This result demonstrates that acAF moderates the accumulation of pathogenic IMMs and neutrophils in the lungs of infected mice, thereby preventing cytokine storm and ALI.

## 4. Discussion

Novel EV-based biologics, derived from tissues or bodily fluids, aim to harness EVs’ natural function as effective cell-to-cell communicators [56,97]. These EV therapies possess unique properties such as non-immunogenicity, internalization through cellular barriers, nano-scale size (<100 nm), and the ability to be manipulated, making them excellent candidates for therapeutic drug and vaccine development [98,99,100]. Vaccine efficiency is based on their ability to deliver wide range of molecules from an area of administration to nearby or long-distance targets in the host body. Due to their composition and characteristics bearing similarities to the parent cell, EVs have been clinically used as therapies against infectious diseases, autoimmune disorders, etc. [101,102,103,104]. EVs are known to interact with various viral pathogens including RNA viruses, such as human immunodeficiency virus 1, hepatitis C virus, Dengue virus, Vesicular Stomatitis virus, influenza A virus, respiratory syncytial virus, and SARS coronavirus, and have been exploited as an anti-viral or preventative measure to limit the spread of viral infections [105,106,107,108,109,110]. Here, we demonstrate the protective role of a human acellular amniotic fluid (acAF) biologic containing amniotic fluid-derived EVs (hAF-EVs) as a therapeutic against SARS-CoV-2 infection, both in vitro and in vivo, using human airway epithelial (A549-hACE2)/mouse macrophage (RAW 264.7) cell lines and a SARS-CoV-2 mouse model, respectively. To our knowledge, this is the first study to explore the novel anti-inflammatory and anti-viral effect of acAF against SARS-CoV-2 infection both in vitro and in vivo.

Timely therapeutic intervention is critical to reduce the progression of severe pneumonia in SARS-CoV-2-infected patients. Treatment with antivirals in the early stages of infection is necessary to ameliorate deteriorating health; however, delayed antiviral therapy is ineffective at reducing disease severity [36,111]. Additionally, the side-effects caused by some antivirals and immunomodulators may result in adverse outcomes [47,111,112,113,114,115]. In our study, we found that acAF is a non-toxic, safe, and novel therapeutic that can suppress viral replication both in vitro and in vivo by promoting the anti-viral response against SARS-CoV-2 infection. A proof-of-concept trial study performed by Bellio et al., investigating the safety and potential efficacy of acAF in SARS-CoV-2-infected patients, corroborated our findings [71]. The diminished inflammatory biomarkers (CRP and IL6) in blood collected from patients in the study by Bellio et al. was similar to the decrease in TNF release observed in mouse macrophages stimulated with TLR agonists and recombinant SARS-CoV-2 proteins in our study, indicating that acAF has anti-inflammatory properties (Figure 3A,B). These results are further supported by an experiment conducted by Li et al., where they found that human amniotic epithelial cells (AECs) were non-toxic and suppressed the chemotaxis of macrophages and neutrophils in murine peritoneal exudates, thereby inhibiting the inflammatory response [88]. Furthermore, the administration of the isolated AF-EV fraction of acAF was found to suppress inflammatory lung injury and immune cell infiltration in a mouse model of bronchopulmonary dysplasia [51]. Additionally the AF-EV fraction suppressed the activation and proliferation of immune cells both in the bronchopulmonary dysplasia mouse model and in PHA-stimulated PBMCs in vitro [51,74]. Similarly, in the current study, treatment with the AF-EV fraction of acAF reduced TNF cytokine release from mouse macrophages stimulated with TLR agonists and recombinant SARS-CoV-2 proteins, further demonstrating the anti-inflammatory capability of AF-EV treatment (Figure 3C,D). Therefore, the suppression of SARS-CoV-2-induced inflammatory mediators by acAF-EVs could be a mechanism behind the decrease in the accumulation of lung-infiltrating myeloid cells in MA-CoV-2-infected mice observed in our study, discussed later in this section.

Previous studies have shown that acAF treatment decreases the progression of clinical outcomes in SARS-CoV-2-infected patients [71,116]. The hyper-inflammatory response serves as the major cause for ARDS and subsequent deteriorating health in SARS-CoV-2-infected patients suffering from disease severity [117]. Koizumi et al. investigated human amniotic epithelial cells (AECs) and found the presence of macrophage migration inhibitory factor (MIF), IL-10, and prostaglandin E2 (PGE2), which have the ability to modulate cytokine-producing inflammatory cells [118]. Considering that the EVs used in this investigation were derived from a similar cell source and that the MIF protein has been identified in AF-EV preparations [51], it can be hypothesized that they can be used for inhibiting robust inflammation occurring during SARS-CoV-2 infection. In our mouse model of SARS-CoV-2, we observed a similar decrease in severity as well as recovery from lethal disease following the administration of acAF (Figure 4A,B). Treatment with acAF also reduced viral titers in the lungs of infected mice compared to untreated mice, indicating its characteristics of suppressing viral replication (Figure 4C). Overall, these immunomodulatory and anti-viral properties of acAF were demonstrated both in vitro and in vivo in our experiments. The differences observed in immunomodulatory effects between preparations may be due to the composition of acAF and AF-EVs. acAF has more soluble molecules than AF-EVs that are removed during the AF-EV precipitation process. Therefore, there may be non-exosome-associated molecules in ac-AF that were the active inhibitory agents of the TRL4 and TLR7 induction of TNF. On the other hand, AF-EVs only were able to inhibit the TLR2 activation of TNF, similar to acAF, suggesting that this inhibition may be AF-EV-mediated. A limitation of these studies includes the ultracentrifugation processing of acAF nanoparticles, as this process may yield the precipitation of non-exosome nanoparticle types that could contribute to the AF-EV-mediated effect. Importantly, the fluorescent nanoparticle tracking analysis of acAF-derived nanoparticles did suggest that the majority of the population of nanoparticles are positive for EV marker CD81; however, this does not rule out the possibility of the likely co-precipitation of other uncharacterized factors.

The acAFs in our study are composed of several microRNA (miRNA) and protein cargos. These have been characterized, and the most concentrated miRNA was found to be Let-7b [51]. Let-7b has been found to inhibit pro-inflammatory responses, especially IL-6 and TNF, by suppressing the TLR4/NFkB pathway in a mouse model of sepsis [119]. Xie et al. also found that the let-7 family of miRNAs suppressed excessive inflammation in an in vitro experiment with a TLR4 agonist (LPS) and SARS-CoV-2 structural proteins (spike and membrane) [120]. The mechanism behind IFN induction can also be attributed to the higher expression of Let-7 miRNA in AF. Wu et al. observed a higher expression of Let-7b in A549 cells infected with influenza A virus. Let-7b induced a type-1 interferon response and, consequently, the inhibition of viral replication [121]. In another study, Wang et al. found that let-7 induction is necessary for recovery from severe SARS-CoV-2 infection [122]. Thus, Let-7b, which is found to be highly expressed in AF, can primarily contribute to anti-viral and anti-inflammatory responses, as observed in our study. We also found that TIMP-2 was highly expressed in the acAF samples (Figure 1D). TIMP-2 plays a role in inhibiting matrix metalloproteinases (MMPs) [123,124], which are involved in regulating pro-inflammatory cytokines such as TNF and IL1β [125,126,127]. As such, a higher expression of TIMP-2 can indirectly suppress TNF induction and could be a potential mechanism behind the acAF-mediated anti-inflammatory response observed (Figure 3A,B).

Lung-infiltrating myeloid cells such as monocyte/macrophages and neutrophils play a pivotal role in enhancing inflammation during SARS-CoV-2 infection [4,8,85,89,94,95]. AECs, as well as AF-EVs, express factors such as MIF, which suppresses the chemotactic recruitment of macrophages and neutrophils, thereby dampening the cytokine storm [87,88,128,129]. A significant decrease in levels of lung-infiltrating IMMs (Figure 5C) and neutrophils (Figure 5F) were observed in MA-CoV-2-infected mice treated with acAF in our study compared to the PBS-treated cohorts. The secretion of MIF by acAF could potentially suppress this migration of macrophages and neutrophils into the lungs and thereby ameliorate the robust inflammatory response occurring during SARS-CoV-2 infection [86,88].

EVs have been used in biomedical research for more than a decade for the diagnosis and treatment of various diseases due to their compatibility, precision, and small size [98,99,100]. As such, the composition and concentration of the nanoparticles within them are critical for therapeutic effects [130]. Based on the difference observed in nanoparticle concentration between lot 3 (~4.04 × 10^10^ nanoparticles) and lot 2 (~2.28 × 10^10^ nanoparticles) (Figure 1B), we believe that the nanoparticle concentration played a significant role in increasing the efficacy of lot 3 in reducing disease severity.

In conclusion, we demonstrate here that acAF is a safe biologic that reduces inflammation and the accumulation of inflammatory myeloid cells in the lungs during SARS-CoV-2 infection in a mouse model. Additionally, acAF reduced the SARS-CoV-2 viral load, promoted recovery, and increased the probability of survival from the disease. Future investigations would benefit by evaluating the AF-EV-mediated anti-proliferative factors and the regulation of T- and B-cell subset responses during SARS-CoV-2 infection to further define their precise mechanism of action.

## Figures and Tables

**Figure 2 viruses-16-00273-f002:**
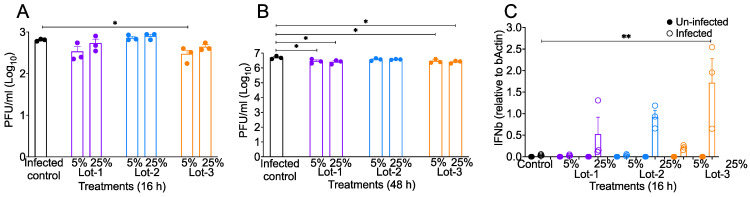
acAF suppresses viral titers and promotes anti-viral response. Supernatants and Trizol-lysed cells were collected from A549-hACE2 cells infected with 0.1 MOI of WT SARS-CoV-2 and treated with 5% and 25% concentrations of acAFs. These were examined for viral titers and anti-viral gene expression. (**A**,**B**) acAF treatment reduced viral replication in A549-hACE2 infected (0.1 MOI of WT SARS-CoV-2) cell supernatants (16 h and 48 h) titered on Vero-hACE2 cells. (**C**) mRNA levels of IFNβ increased in acAF-EV-treated infected cells compared to control cells. Scatter plots show viral plaque-forming units/mL (Log_10_) in cell monolayers per well. Data are representative of 2–3 independent experiments and represented as ±SEM with three technical replicates and two duplicates (**A**,**B**). Each filled circle represents a technical replicate. Statistical significance was determined using one way ANOVA, post hoc Dunnett’s test with * *p* < 0.05, ** *p* < 0.005.

**Figure 3 viruses-16-00273-f003:**
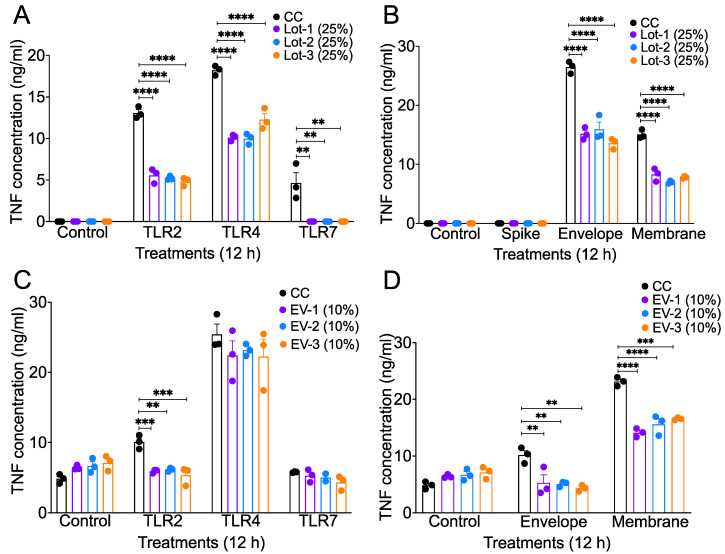
acAF and AF-EVs suppress inflammatory cytokine production by TLR agonist- and SARS-CoV-2 structural protein-stimulated mouse macrophages. Supernatants were collected from RAW 264.7 un-stimulated cells and cells stimulated with TLR agonists (2, 4, and 7) and recombinant SARS-CoV-2 structural proteins (spike—S, envelope—E, and membrane—M) and treated with 25% and 10% acAF and AF-EVs, respectively. These were examined for TNF cytokine production via ELISA. (**A**,**B**) acAF treatment inhibited TNF cytokine levels in cells stimulated with all three TLR agonists and E- and M-recombinant structural proteins compared to TLR agonist- and recombinant protein-only-stimulated control cells (CCs) by more than 2-fold. (**C**,**D**) AF-EV treatment inhibited TNF levels in TLR2, E-, and M-protein-stimulated cells compared to TLR agonist- and recombinant protein-only-stimulated control cells by approximately 1.5- to 2-fold. Data are representative of 2–3 independent experiments and represented as ±SEM with three technical replicates and two duplicates (**A**–**D**). Statistical significance was determined using one way ANOVA, post hoc Dunnett’s test with ** *p* < 0.005, *** *p* < 0.001, and **** *p* < 0.0001.

**Figure 4 viruses-16-00273-f004:**
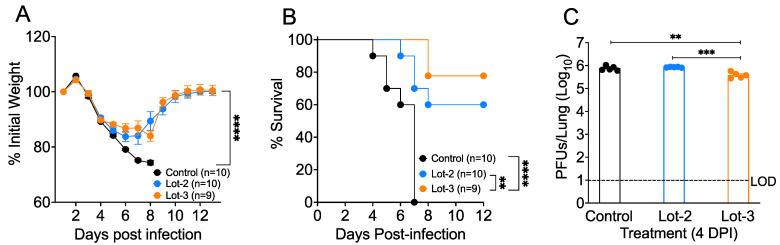
acAF protects against SARS-CoV-2-induced lethal disease and mortality: 10–15-week-old female BALB/C mice treated with acAFs and infected with 250 PFUs/lung of MA-CoV-2 were studied for morbidity, mortality, and viral replication in lungs. (**A**) Observed body weight percent change compared to the baseline measurement over 12 days post infection, (**B**) Percentage of mice survival over the 12-days post infection (**C**) Lung viral titers in the 3 groups at 4 days post infection. Data are pooled from 2 independent experiments with 4–5 mice/group/experiment (**A**,**B**) and are representative of 2 independent experiments with 5 mice/group/experiment (**C**), represented as ±SEM. Statistical significance was determined using one way ANOVA, post hoc (**A**) Dunnett’s test, (**B**) Mantel–Cox test, and (**C**) Tukey’s test with ** *p* < 0.005, *** *p* < 0.001, and **** *p* < 0.0001.

**Figure 5 viruses-16-00273-f005:**
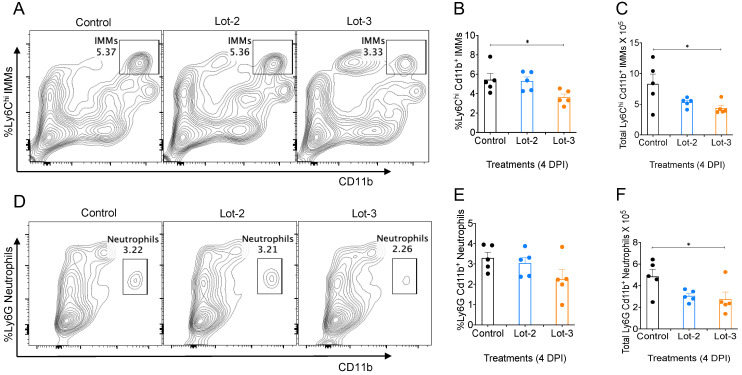
acAF suppresses myeloid cell accumulation in mouse lungs. Lung tissues harvested at 4 DPI from 10–15-week-old female BALB/C mice treated with acAF and infected with 250 PFUs/mouse of MA-CoV-2 were analyzed for lung-infiltrating inflammatory monocyte macrophages (IMMs) and neutrophils. (**A**) Representative FACS plots, (**B**) percentage, and (**C**) total CD11b^+^Ly6C^hi^ IMMs in the lungs of control and acAF-treated mice infected with MA-CoV-2. (**D**) Representative FACS plots, (**E**) percentage, and (**F**) total CD11b^+^Ly6G neutrophils in the lungs of control and acAF-treated mice infected with MA-CoV-2. (**C**,**F**) A 2-fold reduction in the levels of total IMM and neutrophil accumulation was observed in lot 3 acAF-treated mice lungs compared to control and lot 2-treated mice. Data are representative of 2 independent experiments with 5 mice/group/experiment (**C**,**F**) and represented as ±SEM. Statistical significance was determined using one way ANOVA, post hoc Dunnet’s test with * *p* < 0.05.

## Data Availability

Raw and analyzed data will be made available upon request.

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
