# Peer review of "Acellular Human Amniotic Fluid-Derived Extracellular Vesicles as Novel Anti-Inflammatory Therapeutics against SARS-CoV-2 Infection"

_viruses, 2024, doi:10.3390/v16020273_

Round 1

Reviewer 1 Report

Comments and Suggestions for Authors

The manuscript "Human Amniotic Fluid Derived Acellular Extracellular Vesicles 2 as Novel Anti-Inflammatory Therapeutics against SARS-CoV-2 3 Infection" by Chanda et al explored the potential of human AF-EVs in treating SARS-CoV-2 infection and demonstrated its benefits in both in vitro and in vivo models. The project is of novelty and the paper is in a well organized shape. Clarifying the following points will improve the significance and translational potency of the data.

1 Three lots of AFs showed different anti-viral/inflammation efficacies and the authors attributed this difference to the concentrations of nanoparticles in AFs. However, the contributions of different compositions can not be excluded. The best way to clarify this question is to illustrate the compositions of theses AFs by LC-MS or similar methodology. If it is difficult, the authors can at least compare the therapeutic outcomes of AF-EVs, which can be better quantified, instead of AFs in animal models. Another advantage of using AF-EVs is to reduce the possible side effects caused by complex compositions in AFs.

2 Another concerns about the difference in efficacies of different lots is the condition of donors. It is known that virus-specific IgG may be vertically transmitted across placental barrier if the mother got SARS-CoV-2 infection during/before pregnancy. The potential existence of SARS-CoV-2-specific Abs in the AFs need to be assayed/excluded.

3 In Fig. 3 A and C, AFs and AF-EVs of lot 2 and 3 showed different effects on TNF induction. A discussion on this difference is necessary. In addition, the discussion on the potential mechanisms contributing to the IFN induction by AFs are suggested.

Minor issues:

1 In Introduction, the authors may want to update the timeline (replace "3 years ago" in Line 34 with "in 2019").

2 The preparation of AF-EVs is missing in Methods.

3 In Fig. 5A and D, lot 37 and 65 were used to label AFs-treated lungs, whereas only lot 2 and 3 data were summarized in other panels. The authors need to make it consistent.

Author Response

Please attached word document with response to reviewer 1

Reviewer 2 Report

Comments and Suggestions for Authors

Dr. Chanda and colleagues present a study focusing on the immunomodulatory features of EVs derived from amniotic fluid. I found the discussed area interesting as well as having multiple uses in the immunological science. 

After carefully reviewing the draft, I have one major and few minors to be addressed during the peer-review process. 

Major:

Since the Authors focused on EVs from amniotic fluid, it is necessary to provide the methodology for EVs extraction from the fluid. The fluid per se contains lots of proteins, cellular debris, ions etc etc. The proper and validated method for their isolation/purification/densification (if needed) is crucial to validate the conclusions of the suggested findings. Also, some data showing the purity of isolated EVs would be a good addition to the paper. 

Minors:

- The discussion section is very long and redundant at some points. It would be good to shorten it. 

- Explanation for choosing BALB mice should be provided

- lines 498-502 should have the same fonts type and size as the rest of the paper.

- Figure 1 should be moved to the results section, not Materials&Methods.

- Graphs 5A and 5D should have labeled X/Y axes. 

- Do the Authors also consider the role of TLR9 as a player in this mechanism?

Reviewer 3 Report

Comments and Suggestions for Authors

The manuscript explores the therapeutic potential of acellular human amniotic fluid (acAF) containing extracellular vesicles (hAF-EVs) against SARS-CoV-2 infection. The study reveals that acAF is non-toxic and demonstrates significant antiviral properties by reducing SARS-CoV-2 viral titers in human airway epithelial cells. Additionally, acAF effectively suppresses the production of inflammatory cytokines in mouse macrophages, indicating its strong immunomodulatory capabilities. In a mouse model of SARS-CoV-2 infection, acAF treatment resulted in reduced morbidity and mortality, highlighted by decreased body weight loss, lower lung viral titers, and increased survival rates. Furthermore, acAF was found to significantly reduce the accumulation of inflammatory myeloid cells in the lungs, potentially preventing cytokine storm and acute lung injury. These findings collectively suggest that acAF, with its hAF-EVs, is a promising, non-toxic therapeutic agent for managing severe COVID-19 cases by simultaneously suppressing viral replication and the inflammatory response. Overall, this is a very interesting study to be further considered by the journal. I have a few comments below hoping the authors can address.

1.      While the data regarding efficacy is very promising, the manuscript lacks a detailed explanation of the mechanism by which acAF exerts its effects. Including additional data on mechanisms and/or relevant discussions would significantly enhance the overall quality of this work. For instance, the authors could explore and discuss the specific molecular pathways influenced by hAF-EVs through various molecular assays, such as RNA sequencing, qPCR, etc.

2.      Expanding the range of hACE2-expressing cell lines used in in vitro experiments would provide a more comprehensive understanding of acAF’s effectiveness across different cell types.

3.      I am curious about any long-term impacts of the acAF treatment on the treated mice. It would be beneficial if the authors could provide comments or insights on this aspect.

Author Response

Please see attached word document

Round 2

Reviewer 2 Report

Comments and Suggestions for Authors

I was provided with the responses to the Reviewer 3. The attached file doesn't discuss any of my comments/suggestions/questions. Please see below pasted. Until I get revision done accordingly to my comments and discussing my review, I am not able to move review process to the next stage,

Round 3

Reviewer 2 Report

Comments and Suggestions for Authors

The Authors resubmitted their paper entitled Human Amniotic Fluid Derived Acellular Extracellular Vesicles as Novel Anti-Inflammatory Therapeutics against SARS-CoV-2 Infection along with their responses to my former comments.

The Authors addressed most of them in a satisfactory manner, thus I am supportive of the publication once one more thing is fixed:

The Authors used basic method for EVs isolation instead of using size exclusion chromatography columns. It should be acknowledged in the limitation section since we don't know what the exact purity of the isolated EVs is. 

After these 1-2 lines are introduced, I am leaning towards the acceptance of the paper. 
